# Effect of an Electromagnetic Field on Anaerobic Digestion: Comparing an Electromagnetic System (ES), a Microbial Electrolysis System (MEC), and a Control with No External Force

**DOI:** 10.3390/molecules27113372

**Published:** 2022-05-24

**Authors:** Nhlanganiso Ivan Madondo, Emmanuel Kweinor Tetteh, Sudesh Rathilal, Babatunde Femi Bakare

**Affiliations:** 1Green Engineering Research Group, Department of Chemical Engineering, Faculty of Engineering and The Built Environment, Steve Biko Campus, Durban University of Technology, S4 Level 1, Durban 4000, South Africa; emmanuelk@dut.ac.za (E.K.T.); rathilals@dut.ac.za (S.R.); 2Department of Chemical Engineering, Faculty of Engineering, Mangosuthu University of Technology, P.O. Box 12363, Durban 4026, South Africa; bfemi@mut.ac.za

**Keywords:** electromagnetic field, electromagnetic system (ES), microbial electrolysis cell (MEC), anaerobic digestion, methane

## Abstract

This study examined the application of an electromagnetic field to anaerobic digestion by using an electromagnetic system (ES), a microbial electrolysis cell (MEC), and a control with no external force. The experimental work was performed by carrying out biochemical methane potential (BMP) tests using 1 L biodigesters. The bioelectrochemical digesters were supplied with 0.4 V for 30 days at 40 °C. The electromagnetic field of the ES was generated by coiling copper wire to form a solenoid in the BMP system, whereas the MEC consisted of zinc and copper electrodes inside the BMP system. The best performing system was the MEC, with a yield of 292.6 mL CH_4_/g chemical oxygen demand removed (COD_removed_), methane content of 86%, a maximum current density of 23.3 mA/m^2^, a coulombic efficiency of 110.4%, and an electrical conductivity of 180 µS/cm. Above 75% removal of total suspended solids (TSS), total organic carbon (TOC), phosphate, and ammonia nitrogen (NH_3_-N) was also recorded. However, a longer exposure (>8 days) to higher magnetic intensity (6.24 mT) on the ES reduced its overall performance. In terms of energy, the MEC produced the greatest annual energy profit (327.0 ZAR/kWh or 23.36 USD/kWh). The application of an electromagnetic field in anaerobic digestion, especially a MEC, has the potential to maximize the methane production and the degradability of the wastewater organic content.

## 1. Introduction

Globally, the over-reliance on fossil fuels, the formation of excessive amounts of gas emissions, and the rise in fossil fuel prices have encouraged researchers to look for alternative sources of energy that are ecofriendly [1,2,3,4]. In Africa, South Africa is the fifth-most populated country, with a population of almost 59.6 million [5]. The country’s population has grown by almost 16.3 million since 1994. The rapidly growing population and the rapid economic growth and urbanization have led to the production of an enormous amount of wastewater with high solid contents, which is discharged into landfills, surface water, or seawater and, as a result, poses a great threat to the environment and human health and ecological risks [6,7,8,9]. The high population rate and urbanization have also increased the electricity demand. South Africa’s number one electricity producer, in the past few years, has failed to produce enough electricity for the country. With the population constantly increasing, the electricity provider is likely to experience more problems. The electricity provider also faces another issue: approximately 90% of its electricity is generated using coal, which, after being burned, forms greenhouse gas emissions that contaminate the environment and result in environmental temperature changes [10]. Even though South Africa faces these issues, the anaerobic process is the key to decreasing the amounts of organic solids in the ever-increasing effluents produced in South Africa while offering an alternate electricity source that is renewable and ecofriendly. This migration will help address the current challenges faced by the water and energy sector.

The anaerobic process has been used in polluted domestic and industrial wastewater for over a century with the main aim of reducing the amount of biochemical material in the wastewater [11,12]. Anaerobic digestion is a process that involves microbiological, chemical, and physical reactions, where the product of each stage is used as feed in the subsequent stage. The four phases found in anaerobic digestion are hydrolysis, acidogenesis, acetogenesis, and methanogenesis [13]. Hydrolysis is the rate-limiting stage, where organic substrates, such as carbohydrates, lipids, and proteins, are first suspended in wastewater and then broken down by hydrolysis microorganisms into saccharides, long-chain fatty acids, and amino acids, respectively [14,15]. The hydrolysis products are further deteriorated in the acidogenesis stage by microorganisms to generate volatile fatty acids, acetate, hydrogen, alcohols, ammonia, and aldehydes. The acetogenesis stage converts carbon dioxide and organic acids into acetate. The last stage of anaerobic digestion is the methanogenesis stage, in which methanogenesis microorganisms convert intermediate products (such as acetate and volatile fatty acids) into biogas [16]. Biogas usually consists of methane (60%) and carbon dioxide (40%) and may be used to produce energy [17,18].

Currently, researchers are trying to improve the performance of the traditional anaerobic digestion process, mainly because the process has low contaminant removal levels, low microbial activity, and low methane production. The use of an electromagnetic field together with the anaerobic digestion process has been shown to have a great deal of potential due to the greater enhancement of bacterial activity and the higher methane production [19,20]. Interestingly, current studies have shown a more positive effect for the use of a bioelectrochemical system in anaerobic digestion; the use of a bioelectrochemical system enhances the biological activity as well as the electrochemical efficiencies. Even so, the effect of a bioelectrochemical system on the metabolism of microorganisms still requires thorough research, especially since the electrochemical behavior of the ions/protons (i.e., the electrochemical efficiency) in the system is not fully understood. With high electrochemical efficiencies (e.g., coulombic efficiency and current density) possibly resulting in high methane production, very few studies have investigated the path the methane takes in the bioelectrochemical system, i.e., the autotrophic path or the heterotrophic path.

Bioelectrochemical systems (the MFC and the MEC) and electromagnetic systems are the most-used electromagnetic field systems in anaerobic digestion [21,22]. Madondo et al. [23] reported that the most promising bioelectrochemical system is the MEC with 79.1% CH4 removal and 91.6% COD removal from sewage sludge. The MEC was also better in terms of electrochemical measurements, with a maximum current density of 23.3 mA/m^2^, an electrochemical methane yield of 153.44%, and a heterotrophic methane yield of 123.48%, and electrical conductivity of 269.7 µS/cm. This was operated at 40 °C for 25 days. On the other hand, electromagnetic systems (ESs) involve the use of a conductor, such as a wire, and current passes along the conductor to generate a magnetic field. An example of an ES is a solenoid [24]. However, there are few studies on electromagnetism in the anaerobic digestion process. No study has been conducted on the use of an electromagnetic device, such as a solenoid, in the anaerobic digestion of complex matter, e.g., sewage sludge.

More recently, it has been shown that the magnetic field affects several fluid properties, namely viscosity, surface tension, electric charge, and polarization. In the investigations performed to date, electromagnetic fields have mostly been employed for the separation of solids from wastewater, such as waste-activated sludge [25,26]. The impact of electromagnetic fields on electrode kinetics is regarded as the most debatable at present [27]. Since magnetic fields can be achieved in magneto-impedance experiments, Chopart et al. [28] suggested that the magnetic field does not have a substantial influence on kinetics. However, some studies have found that higher magnetic fields result in a blockage of the current density [29]. Factors such as the type of magnetic field, the intensity, the exposure time, and the cell wall can either improve or reduce the performance of the anaerobic digestion process [19]. The magnetic field affects the cell structure transport mechanism of Gram-negative and Gram-positive microorganisms in different ways [30]. Nonetheless, the impact of electromagnetic fields on biochemical processes has been inadequately investigated [31].

In this study, the application of an electromagnetic field in the anaerobic digestion process of sewage sludge was examined by comparing an ES, an MEC, and a control digester with no electromagnetic fields employed in it. The study focused on cumulative biogas generation, the methane composition, electrochemical efficiencies and properties, the stability indicator, and decontamination of the wastewater.

## 2. Results and Discussion

### 2.1. Cumulative Biogas Production and Methane Composition

The cumulative biogas production is significant in anaerobic digestion since it represents the growth rate of microorganisms. Figure 1 shows the cumulative biogas production of the MEC, the ES, and the control for the entire period of the anaerobic process. It was evident that the cumulative biogas production was slow at the beginning of the period for the MEC (< day 2), ES (< day 2), and control (< day 3) digesters. After day 3, there was a high rate of biogas accumulation. However, after day 4, the methanogen growth rate in the ES digester suddenly decreased, and the biogas production of the MEC surpassed that of the ES after day 8. After the sudden increase, biogas production decreased. In the end, the digesters stabilized as they reached the asymptotic phase; the MEC and the control stabilized more quickly than the ES.

The phases of the growth rate are more evident in the daily biogas production graph that is shown in Figure 2. Generally, the rate of biogas generation in anaerobic digestion is directly proportional to the methanogen growth rate [32]. A significant lag phase of microbial growth was evident for all digesters during the initial stage; in a lag phase, adaptation takes place and microorganisms increase in size but not in quantity, which is indicated by low biogas production [33,34]. After passing through the lag phase (day 3), the digesters reached the exponential phase, and the biogas production increased substantially as a result of the exponential growth of methanogenic microorganisms, with the ES showing the highest maximum growth rate [34]. All digesters except the ES reached the stabilized stage after passing through the exponential phase. The digesters then reached the dead phase, which is characterized by an exponential decrease in biogas production.

Table 1 shows the content of methane in the biogas as well as the yield of methane at a hydraulic retention time of 30 days. The MEC had the highest content of methane (86.0%), which was 47.9% higher than the content of methane in the control (38.1%). The biogas yield was also higher for the MEC, with a yield of 404.4 mL/g volatile solids fed (VS_fed_). The yield of methane computed as CH_4_ accumulation per gram of COD removed was 199.3 mL CH_4_/g COD_removed_ for the ES, whereas the yield was 292.6 mL CH_4_/g COD_removed_ for the MEC. The yield for the MEC was more than 3.07 times that for the control (95.3 CH_4_/g COD_removed_). In bioelectrochemical systems, the yield of methane relies on the electrochemically active bacteria and planktonic anaerobic bacteria that contributed to the total CH_4_ generation [35]. Therefore, the high methane content in the MEC was due to the higher numbers of electrochemically active bacteria and planktonic anaerobic bacteria in the system. Nonetheless, a comprehensive study has to be carried out on the electrochemical efficiencies to verify this.

### 2.2. Kinetic Study

The modified Gompertz model was used to determine the influence of inhibition of bacterial activities. The cumulative biogas yields of the MEC, ES, and control were fitted to the modified Gompertz models as depicted in Figure 3. As is evident from the figure, all kinetic models were able to fit the cumulative biogas yield by expressing all three phases of growth, namely the lag phase, the exponential growth phase, and the stabilized phase. The correlation of determination (R^2^) was reasonably high; the R^2^ for the MEC, the ES, and the control was 0.995, 0.963, and 0.987, respectively. The use of the MEC (i.e., zinc and copper electrodes) best fitted the model as it showed the highest R^2^ value. The model with the lowest R^2^ value was the ES, which suggests that this digester had the most uneven biogas yields and, as a result, the modified Gompertz model did not fit the data well. Using the obtained models, the lag phases (λ) were 2.3, 2.2, and 3.1/day, respectively, for the MEC, the ES, and the control. These values correspond to the values obtained using the actual observational data points. The rate constants (k) were 0.35, 0.30, and 0.026/day, respectively. The results suggest that biogas generation and a faster rate of degradation were accomplished at higher rate constant values.

### 2.3. Electrochemical Efficiencies

The electrochemical properties and efficiencies, such as the magnetic field, the electric current, the current density, the power density, the heterotrophic methane yield, the electrochemical methane yield, the coulomb efficiency, and the electrical conductivity, are very important tools in electrochemical systems since they help us understand the way ions behave in the anaerobic system. The magnetic field strength and methane content in the biogas concerning digester type are depicted in Table 2. The ES had the highest magnetic field strength (6.24 mT). It was observed that the higher intensity of the ES (6.24 mT) reduced the methane content in the biogas. An electromagnetic system (ES), such as a solenoid, generates both an electric field and a magnetic field, which contribute to the Lorentz/electromagnetic force as described in Equation (1) [36]:(1)$$F=q\left(E+v\times B\right)$$
where *F* is the Lorentz force, *q* represents the electric charge, *E* represents the electric field due to external flow, *v* represents the velocity, and *B* indicates the magnetic field. Therefore, the combination of the magnetic and electric fields resulted in a high Lorentz force, which, as a result, reduced the performance of the ES. On the other hand, the low magnetic field intensity of the MEC (4.10 mT) had a positive effect on the anaerobic digestion process as it generated a higher methane percentage (86.0%). The absence of a magnetic field in the control made the digester generate the lowest methane content in the biogas (38.1%).

Perhaps one of the most significant electrochemical parameters in electromagnetic field design is current density [37]. Figure 4 shows the current density for the entire digestion period. The magnetic field of the ES performed better before day 8, with higher current density values than that of the MEC. This is because the use of a magnetic field can enhance the way the system performs physically to solid–liquid separation due to the clustering of colloidal particles. Additionally, the use of magnetic fields can affect the biological properties due to the enhancement of the bacterial activity, which increased the current density of the ES [19,38]. However, the higher intensity and the longer exposure duration after day 8 reduced the current density of the ES, which consequently reduced the biogas accumulation (Figure 1). This study affirms the report by Zielinski et al. [24] stating that a longer retention time in electromagnetic field areas can drastically reduce the efficiency of anaerobic digestion. Consequently, magnetic fields can significantly influence methanogenic activities and their microbial community shift. Here, in a low magnetic field (4.10 mT), the MEC stabilized (Figure 3) much more quickly after day 11 at a higher maximum current density (23.3 mA/m^2^) in comparison with the ES, which stabilized after day 14 at a lower maximum current density (17.5 mA/m^2^). This suggests that the adaptability and attachment of the electroactive bacteria in the MEC system were much faster than those of the solenoid in the ES system. Nevertheless, the current densities of both bioelectrochemical systems decreased after 18 days and then stabilized after 22 days, indicating a gradual decline in methane production.

Although it is obvious from this investigation that the use of an electromagnetic field has an effect on the current density and hence methane generation, little is known about the path that the methane takes, i.e., the autotrophic path or the heterotrophic path. The autotrophic (hydrogenotrophic) path involves the formation of complex compounds from simpler substances (2), whereas the heterotrophic (acetoclastic) path involves the breaking down of complex compounds to simpler substances (3) [39].
(2)$$4H_{2}+{\mathrm{CO}}_{2}\to {\mathrm{CH}}_{4}+2H_{2}O\;\Delta G=-135\;\mathrm{kJ}/\mathrm{mol}$$
(3)$${\mathrm{CH}}_{3}\mathrm{COOH}\to {\mathrm{CH}}_{4}+{\mathrm{CO}}_{2}\;\Delta G=-33\;\mathrm{kJ}/\mathrm{mol}$$

The electrochemical methane yield and heterotrophic methane yield are significant electrochemical efficiencies as they help us determine whether the generated methane took the autotrophic path or the heterotrophic path (Figure 5). A system in which both heterotrophic and electrochemical methanogenesis occurs gives rise to a higher rate of conversion of COD to methane, and this occurs at an electrochemical methane yield above 100% [40]. Both bioelectrochemical digesters (the MEC and the ES) generated high electrochemical methane yields; however, the MEC had the highest electrochemical methane yield value (391.6%), implying that a larger percentage of methane was generated through heterotrophic methanogens [40]. The heterotrophic methane yield of the MEC was revealed to be 432.5% of the COD reacted, further indicating a higher rate of carbon dioxide conversion to methane.

Figure 6 depicts the daily electrochemical methane yield and current generation with respect to digester type. It was observed that before day 8, an increase in the current was followed by an increase in the electrochemical methane yield. This suggests that the highest COD to methane conversion rate (i.e., the highest methane generation) occurred during this period. From day 8 to day 19, there was an indirect relationship between the current and the electrochemical methane yield, suggesting that there was a decline in methane generation. After day 19, both the current and the electrochemical methane yield stabilized, indicating that the methane generation had ceased.

Another useful electrochemical indicator that correlates electrochemical methane yield to heterotrophic methane yield is coulombic efficiency, which is used to measure the total electron recovery. This indicator measures the degree to which the electrons generated are transformed into the required product [21]. The higher the coulombic efficiency, the more efficient the bioelectrochemical system [41]. Figure 7 shows the coulombic efficiencies for the MEC and the ES. The MEC digester had a higher coulombic efficiency (110.4%) than the ES digester (105.7%) and this was mostly due to the high methane production (86.0%) [42,43]. Furthermore, the coulombic efficiency depends on the system’s resistance (or impedance), and higher resistance in the system leads to a decrease in the coulombic efficiency since it is not easy to retrieve electrons from a system with higher resistance [44]. Therefore, the higher coulombic efficiency of the MEC implies a lower resistance (ohmic losses) and hence a more efficient system. This shows that more electrons were transformed into heterotrophic methane in the MEC as compared with the ES.

Electrical conductivity is another effective electrochemical property that can be used to identify the electrical flow inside the reactor and not outside the reactor or external circuit. The maximum current density and electrical conductivity with respect to digester type are depicted in Figure 7. The results show that a decrease in the current density led to a decrease in electrical conductivity. The same conclusion was drawn by the authors of [45], who did a study on the influence of magnetic fields on gas bubbles and found that current density was proportional to electrical conductivity. Increasing such MEC characteristics as the current density and coulombic efficiency not only increases the electrical conductivity of the solution but also lowers the impedance between the anode and the cathode [22,46]. The MEC generated the highest electrical conductivity (180 µS/cm), which was greater than the ES value of 151.3 µS/cm. Therefore, the high current density in the MEC digester certainly enhanced the performance of the MEC by decreasing the impedance that hinders the flow of electrons and ions and, as a result, increased the electrical conductivity for enhanced methane generation.

### 2.4. Process Stability Indicator

Despite the fact that the results on the electrochemical efficiencies reveal that the magnetic field, current density, power density, electrochemical methane yield, heterotrophic methane yield, coulomb efficiency, and electrical conductivity had an impact on the methane production, it is essential to find out whether protons and ions influenced the pH of the ES and MEC. The microbial activity depends on the pH; microorganisms are highly sensitive to the pH of the system, and variations in the pH could result in variations in the respiration of microorganisms and, as a result, extracellular electron transfer [22]. Furthermore, parameters such as electrical conductivity, coulombic efficiency, ion transfer, and oxidation of a substrate are indirectly or directly linked to pH [22]. It is evident from Table 3 that an increase in coulombic efficiency resulted in a decrease in pH, with the MEC having the lowest pH (7.1, i.e., a neutral value). The high coulombic efficiency of the MEC (110.4%) lowered the pH of the system. According to [47], low pH values (especially neutral values) improve the performance of the bioelectrochemical system by lowering cathode overpotentials and the solvent resistance, which is why the MEC performed better. Moreover, the combination of a neutral pH and high coulombic efficiency in the MEC is an indication of both electrochemical and heterotrophic methane generation [40].

### 2.5. Influence of the Electromagnetic Field on Decontamination

The water quality parameters that were measured for decontamination are shown in Figure 8 and include COD, total suspended solids (TSS), total organic carbon (TOC), color, phosphate, and ammonia nitrogen (NH_3_-N). It was found that the MEC had the highest removal values for COD (98.6%), TOC (95.9%), TSS (88.4%), phosphate (76.3%), and NH_3_-N (43.7%). This can be attributed to the higher electron recovery efficiencies of the MEC system, namely the maximum current density (23.3 mA/m^2^), coulombic efficiency (110.4%), and electrical conductivity (180 µS/cm) [22,23,48,49]. Conversely, exposure to electromagnetism (the ES) resulted in better removal of color (97.9%). Other researchers have also found that exposure to a higher magnetic field strongly enhances the efficiency of the removal of color from water [50,51].

### 2.6. Economic and Energetic Viability of the Processes

In order to determine the efficiency and economic viability of the bioelectrochemical systems (the MEC and the ES), a techno-economic analysis was performed. All energy-related calculations were based on methane yield. Biogas usually consists of approximately 60% methane and 40% carbon dioxide, and it was assumed that 80% of the energy generated is transformed into electrical power [17,18]. The energy cost was based on the biogas generated by the digesters (MEC = 404.4 mL/day, ES = 330.9 mL/day, and control = 169.1 mL/day). The cost estimation for the digesters is shown in Table 4.

Equation (4) below can be used to determine the energy generated per day (E_G_):(4)$$E_{G}={\dot{Q}}_{{\mathrm{CH}}_{4}}\times {\mathrm{LHV}}_{{\mathrm{CH}}_{4}}$$
where ${\dot{Q}}_{{\mathrm{CH}}_{4}}$ is the rate of CH_4_ generated in m^3^ CH_4_/day, and ${\mathrm{LHV}}_{{\mathrm{CH}}_{4}}$ is the lower heating value, which is 35.8 kJ/m^3^ CH_4_.

The energy needed by the water bath (E_B_) can be estimated by Equation (5):(5)$$E_{B}=\dot{m}\times C_{p}\times \Delta T=\dot{Q}\times \rho \times C_{p}\times \left(T_{1}-T_{0}\right)$$
where $\dot{m}$ (= $\dot{Q}\times \rho$) is the mass flowrate in kg/day, $\dot{Q}$ represents the substrate volumetric flowrate in m^3^/day, $C_{p}$ is the specific heat capacity in kJ/kg·°C, ρ is the density in kg/m^3^, $T_{1}$ is the digester temperature in °C, and $T_{0}$ represents the temperature of the substrate in °C.

The energy used by the system with an external power supply (E_E_) can be obtained by Equation (6):(6)$$E_{E}=I\times V$$
where I is the current in A and V is the voltage in V.

The total energy generated was computed as the difference between the generated energy and the consumed energy (Equation (7)):(7)$$E_{T}=E_{G}-E_{B}-E_{E}$$

In terms of the techno-economic analysis, the systems with an external power supply of 0.4 V (i.e., both the MEC and the ES) were more economical than the control. The most economical digester was the MEC. The MEC had a net energy profit of 327.0 ZAR/kWh (23.36 USD/kWh), which was approximately 11.2 times that of the control (29.20 ZAR/kWh or 2.102 USD/kWh).

## 3. Materials and Methods

### 3.1. Equipment and Operation

The biochemical methane potential (BMP) experiments were performed via three types of biodigesters, namely (1) a microbial electrolysis cell (MEC), (2) an electromagnetic system (ES), and (3) a control with no external force (Figure 9). Each biodigester was a 1 L Duran Schott bottle (Laboratory Supplies Co., Durban, South Africa) and included a working volume of 80% (*v*/*v*) and a headspace of 20% (*v*/*v*). The biodigesters were fed with sewage sludge (0.3 L) and waste activated sludge (0.5 L), which were both acquired from a wastewater treatment company located in Durban, South Africa. The BMP tests were conducted at 40 °C for a duration of 30 days and the organic loading rate was 1.13 g VS/L-d. The MEC digester comprised a zinc electrode (anode) and a copper electrode (cathode), and the electrodes had a length of 12 cm and a width of 1 cm. The MEC was powered with 0.4 V [35,52] using a Matrix MPS-3005S DC power supply. On the other hand, the ES was an electromagnetic digester with a solenoid inside it that was made out of copper (diameter, 0.1 cm; coil length, 150 cm). The electromagnetic field was produced by the solenoid’s DC power supply (0.4 V) and by coiling the copper to form a coil with 58 turns (immersed length, 9 cm; immersed diameter, 0.7 cm). These dimensions were selected to ensure that the immersed area (viz. the surface area per volume) of the ES was similar to that of the MEC digester.

### 3.2. Analytical Parameters and Calculations

The daily biogas volume was obtained using the water displacement method. A Geotech GA 5000 Portable Biogas Analyzer was used to analyze the composition of the biogas. The contaminants that were measured (in the feed and after digestion) were COD, TSS, NH_3_-N, color, and TOC, which were obtained using a Hach DR 3900 calorimeter (Hach, Loveland, Colorado, USA). The percentage of contaminant removed (CR) was obtained by (8):(8)$$\%\;\mathrm{CR}=\frac{\mathrm{influent}\;\mathrm{contaminant}-\mathrm{contaminant}\;\mathrm{after}\;\mathrm{digestion}}{\mathrm{influent}\;\mathrm{contaminant}}\times 100\%$$
where the influent contaminant represents the type of contaminant in the feed, i.e., COD, TSS, color, TOC, NH_3_-N, and phosphate.

The pH of the biodigesters was measured before digestion and after the digestion process using a Hanna H198129 conductivity meter. The electrochemical efficiencies that were measured were the electric current, electric potential, magnetic field, current density, power density, electrochemical methane yield, heterotrophic methane yield, and coulomb efficiency. A FLUKE 177 RMS multimeter was used to measure both the electric current and the electric voltage across the MEC and ES terminals. The magnetic field in the biodigesters was obtained using a digital Telsameter.

The current density (j) of the bioelectrochemical system was determined by (9) [53]:(9)$$j=\frac{I}{A}$$
where I represent the current (Amps) and A is the anode cross-section (m^2^).

The electrochemical methane yield and heterotrophic methane yield were determined by (10) and (11), respectively [40].
(10)$$\mathrm{Electrochemical}\;\mathrm{methane}\;\mathrm{yield}=\frac{\left\lfloor \frac{V_{M}}{22.450}\right\rfloor }{\left[\frac{n_{E}}{8\times F}\right]}\times 100\%$$
(11)$$\mathrm{Heterotrophic}\;\mathrm{methane}\;\mathrm{yield}=\frac{\left\lfloor \frac{V_{M}}{22.450}\right\rfloor }{\left[\frac{\left({\mathrm{COD}}_{\mathrm{feed}}-{\mathrm{COD}}_{\mathrm{digestate}}\right)\times V_{F}}{64}\right]}\times 100\%$$
where $V_{M}$ denotes the methane accumulation in mL, $n_{E}$ denotes the amount of e^−^, which is determined as the region underneath the graph of current against time (i.e., $n_{E}=\int I.dt$), F is Faraday’s constant, which is 96.485 C/mol e^−^, and $V_{F}$ denotes the treated (feed) wastewater volume (mL). The coulombic efficiency (CE) can be obtained by (12) [54]:(12)$$\mathrm{Coulombic}\;\mathrm{efficiency}=\frac{\mathrm{Heterotrophic}\;\mathrm{methane}\;\mathrm{yield}}{\mathrm{Electrochemical}\;\mathrm{methane}\;\mathrm{yield}}\times 100\%=\frac{8\times n_{E}}{F\times V_{F}\times \left({\mathrm{COD}}_{\mathrm{feed}}-{\mathrm{COD}}_{\mathrm{digestate}}\right)}\times 100\%$$

The cumulative biogas yield of the MEC, the ES, and the control was modeled using the modified Gompertz model equation (Equation (13)). The modified Gompertz model can be used to represent the three phases of growth.
(13)$$Y=\mathrm{Aexp}\left\{-\exp\left[\frac{\mu_{m}e}{A}\left(\lambda -t\right)+1\right]\right\}=A\left(1-\exp\left(-\mathrm{kt}\right)\right)$$
where Y is the cumulative specific yield in mL/g VS; $\mu_{m}$ represents the maximum specific growth rate in mL/g VS.d., i.e., the tangent at the inflection point; A denotes the highest value obtained on the *y*-axis in mL/g VS; λ denotes the phase-in days; t is the time in days; e represents 2.71828; and k (= $\frac{\mu_{m}e}{A}$) is the rate constant in methane produced per day.

The physicochemical properties of the sewage sludge and the waste-activated sludge are shown in Table 5.

### 3.3. Statistical Test

Statistical analysis was conducted on the observational data points shown in Figure 1, Figure 2, Figure 4, Figure 6, and Figure 8 with error bars. The error bars represent potential error amounts that are graphically relative to each observational point or data marker in a data series. In all graphs, the error bars denote the standard error of the mean and were added to each graph using the ‘Add Chart Element’ tool in Microsoft Excel. The data presented were based on duplicated experimental setups and samples analyzed in triplicate.

## 4. Conclusions

In this investigation, the influence of an electromagnetic field on anaerobic digestion was examined using an ES and a MEC. Under the experimental conditions employed, the application of an electromagnetic field exerted a significant influence on the anaerobic digestion process. The performance of the MEC in terms of decontamination was better than that of the ES, with higher COD (98.6%), TOC (95.9%), TSS (88.4%), phosphate (76.3%), and NH_3_-N (43.7%) removal values. The MEC also showed higher electron activity as it produced higher electrochemical efficiencies, including electrochemical methane yield (391.6%), heterotrophic methane yield (432.5%), and coulombic efficiency (110.4%), and electrical conductivity (180 µS/cm). As a result of the high electrochemical efficiencies, the methane content in the biogas was improved by 47.9% (>38.1% of the control), and the highest methane yield (292.6 mL CH_4_/g COD_removed_) and biogas yield (404.4 mL/g VS) were generated by the MEC. It also took a shorter period for the MEC to stabilize (11 days) as compared with the ES, which stabilized after 14 days and at a lower current density (17.5 mA/m^2^). In contrast, the longer period of exposure to the magnetic intensity of 6.24 mT on the ES reduced its overall anaerobic digestion performance. The biogas yields of the digesters were fitted to the modified Gompertz model. The best-fitted digester was the MEC, with a coefficient of determination (R^2^) of 0.995. The use of an external power supply revealed a net energy profit of more than 217.1 ZAR/kWh (15.6 USD/kWh) compared with the control, with the MEC producing the highest net energy profit. Because of the present prospects of electromagnetic fields, this investigation affirms the application of the MEC as being promising for the treatment of wastewater and economical as well as green energy generation. The microbial activity and other related operating parameters warrant future research attention as they were not considered in this study.

## Figures and Tables

**Figure 1 molecules-27-03372-f001:**
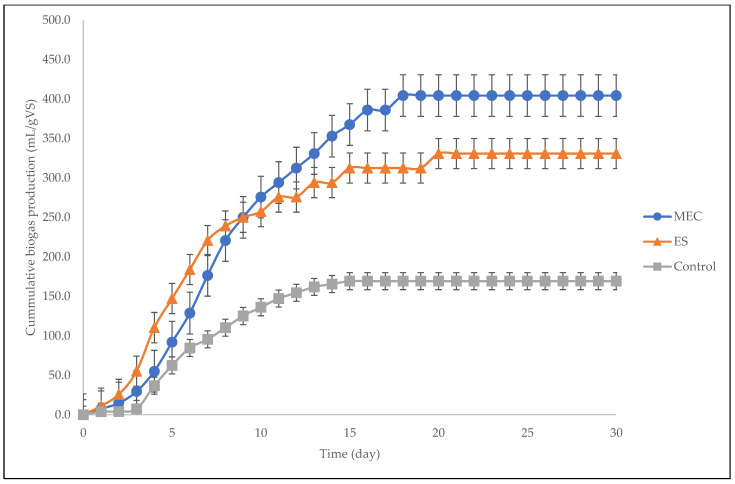
Cumulative biogas production for MEC, ES and control.

**Figure 2 molecules-27-03372-f002:**
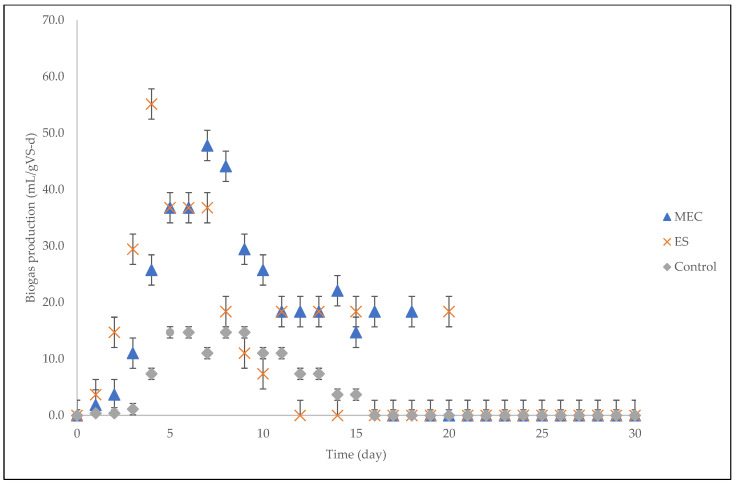
Daily biogas production over a hydraulic retention time of 30 days.

**Figure 3 molecules-27-03372-f003:**
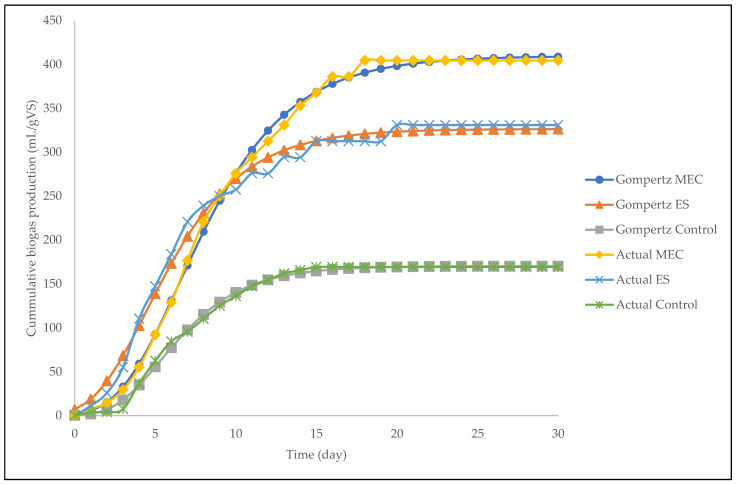
The modified Gompertz models versus actual cumulative biogas production for all digesters.

**Figure 4 molecules-27-03372-f004:**
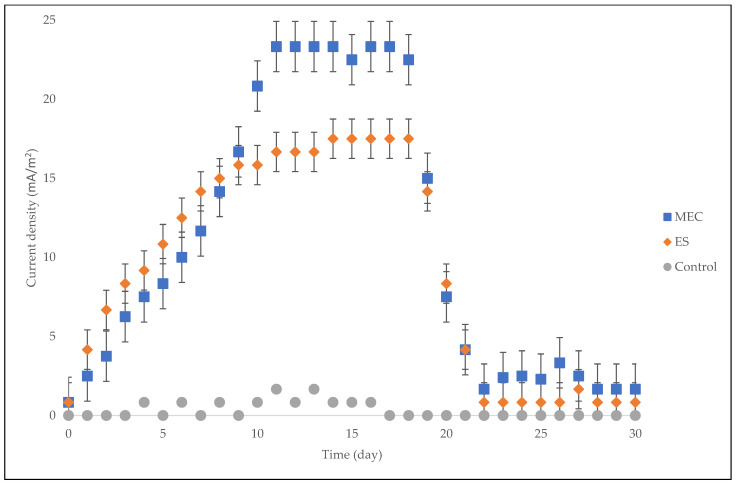
Current density for MEC, ES and control over a period of 30 days.

**Figure 5 molecules-27-03372-f005:**
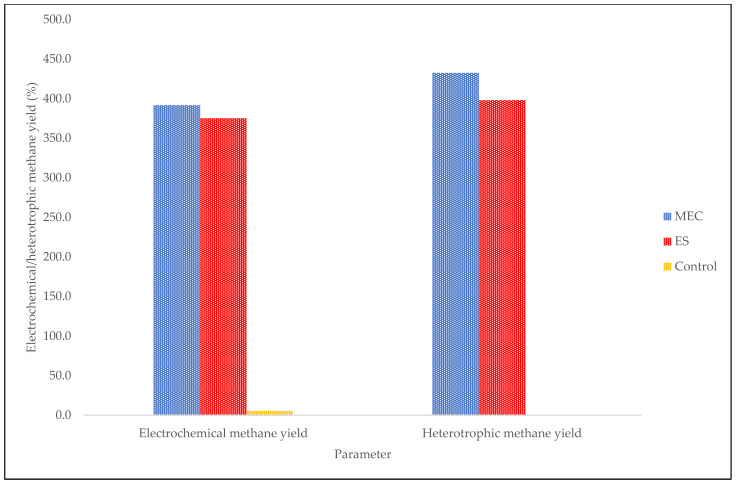
Electrochemical methane yield and heterotrophic methane yield with respect to digester type after the hydraulic retention period of 30 days.

**Figure 6 molecules-27-03372-f006:**
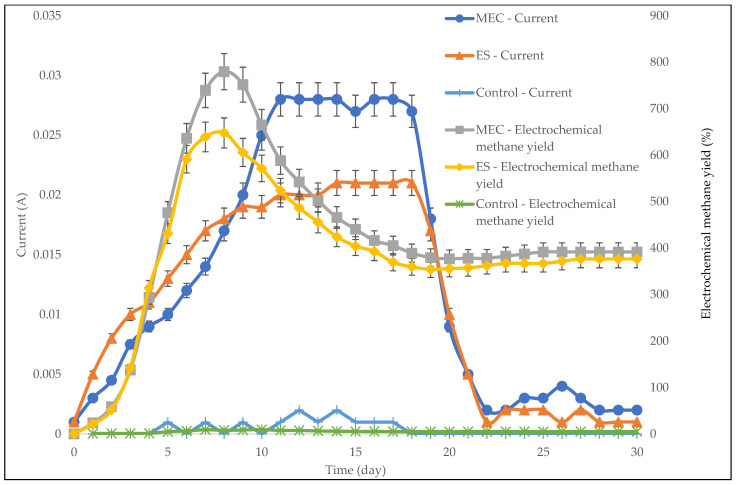
Daily current generation and electrochemical methane yield for the ES and the MEC.

**Figure 7 molecules-27-03372-f007:**
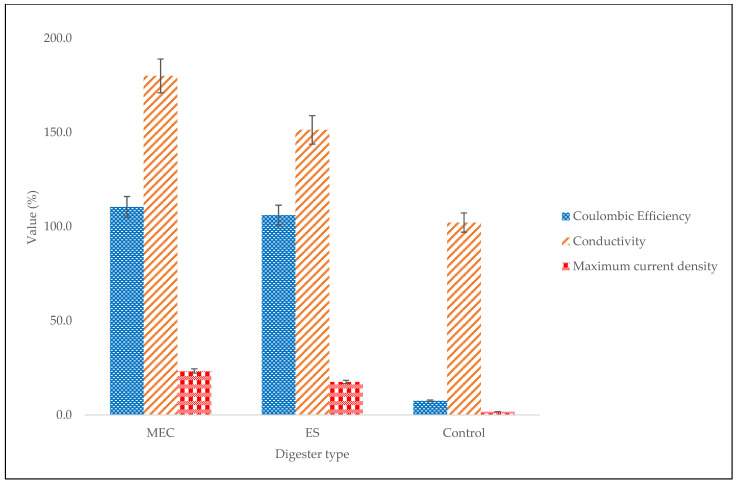
Coulombic efficiency, electrical conductivity, and maximum current density with respect to digester type.

**Figure 8 molecules-27-03372-f008:**
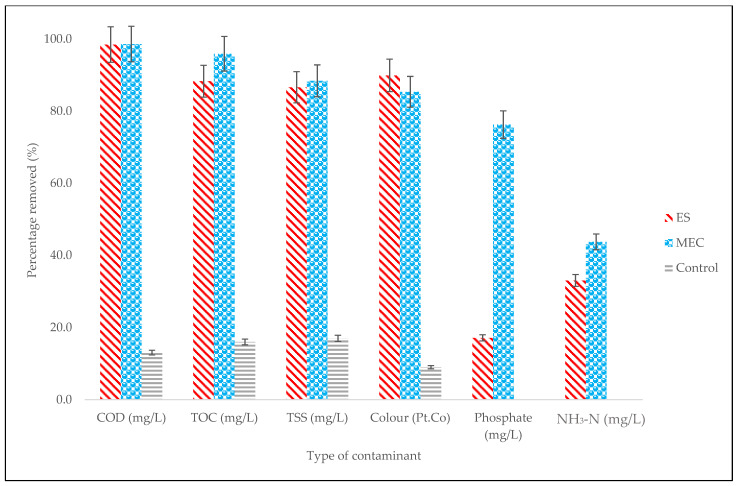
Effect of digester type on decontamination.

**Figure 9 molecules-27-03372-f009:**
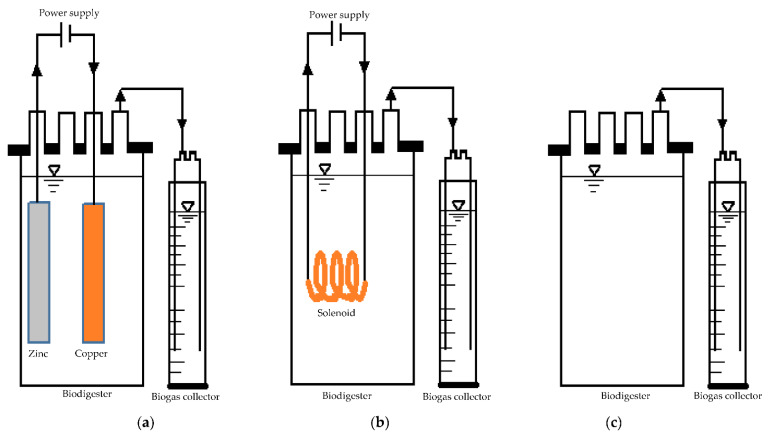
Schematic diagram of (**a**) the microbial electrolysis cell (MEC) with zinc and copper electrodes; (**b**) the electromagnetic system (ES) with a solenoid; and (**c**) the control with no power supply.

**Table 1 molecules-27-03372-t001:** Methane content and methane yield.

Digester Type	Content of CH_4_ (%)	Yield (mL CH_4_/g COD_removed_)	Yield (mL/g VS_fed_)
ES	79.0	199.3	330.9
MEC	86.0	292.6	404.4
Control	38.1	95.3	169.1

**Table 2 molecules-27-03372-t002:** Magnetic field and methane content in the biogas.

Digester Type	Magnetic Field (mT)	CH_4_ (%)
MEC	4.10	86.0
ES	6.24	79.0
Control	0.01	38.1

**Table 3 molecules-27-03372-t003:** Coulombic efficiency and pH with respect to digester type.

Digester Type	Coulombic Efficiency (%)	pH
Control	7.5	7.6
ES	95.3	7.5
MEC	110.4	7.1

**Table 4 molecules-27-03372-t004:** Economic and energetic viability of the processes.

Type	Unit	MEC	ES	Control
Energy content of CH_4_	m^3^/h	0.00362	0.00296	0.00017
Energy generated (E_G_)	kWh	0.00289	0.00237	0.00035
Energy utilized by the water bath (E_B_)	kWh	0.00053	0.00059	0.00014
Energy used by the system with an external power supply (E_E_)	kW/h	0.00001	0.00008	0.00000
Total energy (E_T_)	kWh	0.00235	0.00177	0.00021
Net energy profit per day (3.22 ZAR/kWh)	ZAR/kWh	0.00757	0.00570	0.00068
Net energy profit per day (0.23 USD/kWh)	USD/kWh	0.00054	0.00041	0.00005
Net energy profit per year	ZAR/kWh	327.0	246.3	29.20
Net energy profit per year	USD/kWh	23.36	17.72	2.102

**Table 5 molecules-27-03372-t005:** Physico-chemical properties of the sewage sludge and waste activated sludge.

Parameters	Unit	Sewage	Waste-Activated Sludge
pH	-	7.0 ± 0.96	6.8 ± 0.55
Density	kg/m^3^	1094 ± 25.02	1022 ± 20.21
NH_3_-N	mg/L	41.02 ± 1.89	30.01 ± 2.01
TOC	mg/L	3637.45 ± 46.98	1774.23 ± 39.87
Phosphate	mg/L	11.97 ± 0.12	8.63 ± 0.15
VS	mg/L	45.53 ± 1.31	30.55 ± 1.71
TS	mg/L	52.33 ± 5.04	39.47 ± 4.10
COD	mg/L	4501.54 ± 220	1020 ± 78.1
Color	Pt.Co	435.02 ± 4.23	122.01 ± 2.43

## Data Availability

Data is contained within the article.

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
