# Peer review of "Effect of an Electromagnetic Field on Anaerobic Digestion: Comparing an Electromagnetic System (ES), a Microbial Electrolysis System (MEC), and a Control with No External Force"

_molecules, 2022, doi:10.3390/molecules27113372_

Round 1
Reviewer 1 Report
- A would suggest not to introduce MFC, bioelectrochemical system, MEC altogether. The introduction should only be restricted to MEC and ES.
- The author should check the Y-axis unit in Fig 1. It suggests after 30 days of operation, the methane yield is just 4-5 ml? Moreover, it suggests there is no difference between EC and control digester in terms of methane yield. The author should refrain from smoothing the lines joining the observational points.
- Again, in figure 2, the biogas production is just a few ml? I am sure, there must be some calculation error. The values in Table 1 are also suggesting the same. Yield values (ml CH4/g COD removed) across control and ES are not such different!! (0.95 ml vs. 113 ml)
- Figure 3 is completely non- decipherable. The author wants to show methane yield at two different magnetic field strengths with two digester types. If so, it would have been easy to show in a grouped bar.
- Figures 7,8,9 are a very simple representations. Just two columns, which can be easily described in the text. Else, all these figures can be combined easily.
- What does the error bar represent in Figure 10? standard error of mean? If so, why the error values are high at higher removal efficiency and low at lower removal values? Measurement errors are likely to be normally distributed and should have no bias.
- Setting up an anaerobic digester requires adjustment of alkalinity, creation of an anaerobic environment, and measurement of physical/chemical properties of the substrate, the specific methanogenic activity of seed sludge. That information is missing from the M&M sections.
- In eq. 4, what is referred to as feed contaminant should be clearly explained? Is it COD of feed? In addition, I wonder why turbidity was measured? The system is certainly turbid throughout the AD process.
Author Response
Please find attached file for your reference

Reviewer 2 Report
Manuscript ID molecules-1632687 entitled “Effect of constant magnetic field on anaerobic digestion: com-paring electromagnetism and microbial electrolysis” presents an interesting topic that may be of interest to readers. However, in its current form, the manuscript does not meet the standards of scientific work. In my opinion, it should be rejected and re-submitted after deep corrections. Some of my remarks below:
- The title is not clear and the content is in not accord with title. Authors used (1) a microbial electrolysis cell (MEC), (2) an electromagnetic system (ES), and (3) a control with no external force. Where is constant magnetic field emitted by permanent magnets. Only literature reference? Title have have to be changed and refer only to research work.
- Throughout the manuscript, it is better to use electromagnetic field rather than constant magnetic field because this is confusing.
- The Abstract is clear and refers to the study findings, methodologies, discussion as well as conclusion but the graphical abstract would be highly desirable in order to present the readers with the authors' intentions and the research organization scheme. It needs to be completed.
- The keywords should be changed according to my remarks.
- In the introduction section present good enough background of the researches. The authors must underline the major findings of their work and explain novelty of this study comparatively with their published papers or other similar studies.
- Aim of the study is clear but I suggest use,as above, formulation “electromagnetic field” in relation to own research. Of course, in the discussion, in many places the use of constant magnetic field is justified.
- Please take into account the research hypotheses that were formulated and that were verified.
- The lack of a correctly conducted statistical analysis eliminates the credibility of the results and excludes the correctness of inference. The methodology lacks information on the performed statistical tests verifying the significance of differences between the analyzed variables. What significance/ probability level was used, what post-hoc tests were used?
- Lack of information on the number of replicates of the study.
- Lack of information on the production of biogas and methane by the sludge inoculum without substrate.
- The exact physico-chemical characteristics of the anaerobic sludge inoculum and the substrate (sewage sludge) must be given in the Methodology section.
- Carefully check and correct the legends on all figures.
- As standard for presentation of biogas and methane production is the results are given per unit of dry organic matter. For this reason the knowledge of sludge characteristics is necessary. I needs to be done.
- Organic load rate (OLR) should be presented.
- Please specify the economic and energetic viability of the process. Perform also an energy balance.
- As standard, in methane fermentation respirometric tests, the biogas production reaction rate, the reaction rate constant and the model described by a mathematical equation are presented. It needs to be completed.
- The paper was written in standard, but correction English is necessary. The size of the article is appropriate to the contents.
- The manuscript adheres to the journal's standards after major revision.
Author Response

(The authors gave the same response as above.)

Reviewer 3 Report
- The statement “It was observed that there is an inverse relationship between magnetic field strength and methane content in biogas; the higher intensity of the ES (6.24 mT) reduced the content of methane in biogas” could not be deduced from Figure 3 as there is only two data point of observation. The bar graph did not represent a trend but rather a comparative standpoint.
- The control set-up is excluded in the measurement of electrochemical methane yield, heterotrophic methane yield but it is included in the measurement of other parameters.
- Although methanogens activity or biogas production is directly proportional to the methanogens growth rate, measuring the methanogen population directly can provide a better vantage point. This can establish the time points where their growth is maximum or optimal and if it corresponds to the time points where the methane production is also at its peak.
- There was little or no explanation as to why there is a sudden rise and sudden drop in the daily methane production of ES on days 3-6 as shown in Figure 2. This is where indirectly/directly measuring the bacterial population could have supported this observation. Additionally, as the electrochemical make-up of an environment affects pH, measuring it daily could provide additional data to possibly explain the varied daily behavior of the set-ups.
- The study presents well-supported conclusions as it measured several parameters that are in congruence with one another. However, measuring some variables on a daily basis and including other parameters as well could make it better-rounded.
Author Response

(The authors gave the same response as above.)

Round 2
Reviewer 2 Report
Thanks a lot to the Authors for taking into account my comments and for significantly improving the manuscript. In my opinion, the manuscript can be published in the current form.
Author Response
Thanks very much for your input
Reviewer 3 Report
The paper discussed the effects of different electromagnetic field source on anaerobic digestion. Its main contribution is substantiating previous observations on the effect of external electromagnetic force on biological system-based processes. I recommend for the paper to be accepted with minor comments. The parameters investigated are well-suited with the objectives of the experiments. The discussion could have been enriched if there were portions of the study dedicated to measuring bacterial growth and other parameters related to it –e.g. pH. Nonetheless, they were able to establish the relationship of biogas/methane production to the bacterial growth by dedicating a well-cited and comprehensive discussion about them. In terms of graphical presentation, connected scatterplots seem to be more applicable for graphs that are presenting multiple dataset and trends; as for example figure 2 wherein most data points coincide with one another making it harder to visualize the trend for each set-up.
Author Response
We totally agree with the comments on the study of the bacterial growth and other related parameters would have enriched the manuscript. Unfortunately, the microbial activity and community shift was out of scope of this current study which will be taking into account in future works as a recommendation. Figure 2 and 4 is revised as suggested